# Prognostic Factors Associated with Biochemical Relapse After Radiotherapy in Localized Prostate Cancer: A Retrospective Cohort Study

**DOI:** 10.3390/biomedicines13092185

**Published:** 2025-09-07

**Authors:** Nicolas Feltes Benitez, Felipe Couñago, Saturio Paredes Rubio, Manuel Galdeano-Rubio, Esther Jovell-Fernandez

**Affiliations:** 1Radiation Oncology Department, University Center, Hospital Universitario de Terrassa (CST), Ctra. de Torrebonica, s/n, 08227 Terrassa, Spain; sparedes@cst.cat (S.P.R.); mgaldeano@cst.cat (M.G.-R.); 2Department of Medicine, School of Medicine and Health Sciences, Universitat Internacional de Catalunya (UIC Barcelona), Campus Sant Cugat, Josep Trueta s/n, Sant Cugat del Vallès, 08195 Barcelona, Spain; ejovell@uic.es; 3Department of Medicine, Faculty of Medicine, Health and Sports, European University of Madrid, 28108 Madrid, Spain; felipe.counago@genesiscare.es; 4Department of Radiation Oncology, Hospital San Francisco de Asís and Vithas La Milagrosa, Genesis Care, 28002 Madrid, Spain; 5Epidemiology Department, University Center, Hospital Universitario de Terrassa (CST), Ctra. de Torrebonica, s/n, 08227 Terrassa, Spain

**Keywords:** prostate cancer, external-beam radiotherapy, biochemical recurrence, PSA, positive biopsy cores, perineural invasion, Gleason score/ISUP grade, risk stratification

## Abstract

**Background**: Biochemical recurrence (BCR) after definitive radiotherapy (RT) in localized prostate cancer (PC) is a clinically relevant event that impacts long-term management and prognosis. However, the prognostic value of certain biopsy-derived pathological parameters remains underexplored in RT-treated cohorts. **Methods**: We retrospectively analyzed 444 patients with localized PC treated with external beam radiotherapy (with or without androgen deprivation therapy) between 2013 and 2019. Clinical, radiological, and detailed histopathological data, including Gleason score, perineural invasion, and the number and proportion of positive biopsy cores—were collected. Logistic regression models were used to identify predictors of BCR. **Results**: After a median follow-up of 72 months, 11.7% of patients developed BCR. In multivariable analysis, higher PSA at diagnosis (*p* = 0.05), higher Gleason score (ISUP ≥ 4; *p* = 0.036), and greater tumor burden in biopsy cores—quantified as both the number and proportion of positive cores per lobe and overall (*p* < 0.05)—were independently associated with BCR. Perineural invasion showed a univariable association (*p* = 0.036), though it did not remain significant after adjustment. Overall, 19.2% (10/52) of recurrences were diagnosed beyond five years post-treatment, underscoring the need for prolonged follow-up. **Conclusions**: PSA at diagnosis, the extent of tumor involvement in diagnostic biopsies, and ISUP grade group ≥ 4 (Gleason score ≥ 8) were identified as independent predictors of biochemical recurrence after RT in localized prostate cancer. Lower Gleason categories and perineural invasion showed only a trend toward significance in the multivariable analysis, suggesting that their predictive effect may be attenuated by other covariates.

## 1. Introduction

Prostate cancer (PC) is the second-most frequently diagnosed malignancy among men worldwide. In Spain alone, 32,967 new cases were reported in 2022, according to GLOBOCAN data [1]. Since the introduction of the Gleason score (GS), serum prostate-specific antigen (PSA) testing, and the ability to perform clinical staging through magnetic resonance imaging (MRI), these parameters have been widely used to predict disease progression and to guide therapeutic decision-making—whether surgery, radiotherapy (RT), or active surveillance [2,3].

The combination of these three factors has enabled the identification of validated risk groups based on the likelihood of biochemical recurrence (BCR) [4]. These risk-stratification systems now constitute the backbone of major international clinical guidelines [5,6,7]. In addition to the classic predictors (PSA, clinical stage, and GS), other markers have been proposed as potential predictors of BCR, such as the absolute number as well as the percentage of positive biopsy cores and the presence of perineural invasion (PNI), among others [8,9]. However, most of the available evidence for these factors comes from surgical series, and their prognostic value in patients treated with RT remains less established and has been insufficiently investigated to date [10,11].

External-beam radiotherapy (EBRT), either alone or in combination with androgen-deprivation therapy (ADT), is a well-established and effective treatment modality for localized PC across all risk groups [12,13,14]. Nevertheless, despite modern image-guided and intensity-modulated techniques, a significant proportion of patients treated with curative intent will experience BCR in the years following therapy. It is estimated that 15–35% of cases recur biochemically within the first five years after EBRT [15].

The detection of BCR has important clinical implications, as it frequently precedes overt disease progression. Identifying recurrence early, particularly at PSA levels near the Phoenix threshold (defined as a rise of ≥2 ng/mL above the post-treatment nadir PSA, according to the RTOG-ASTRO Phoenix criteria), can allow for timely intervention during a less advanced stage of tumor evolution [16].

In this context, the high sensitivity and specificity of current imaging techniques—particularly prostate-specific membrane antigen positron-emission tomography (PSMA-PET)—have enabled targeted salvage treatments, such as focal or oligometastasis-directed therapies, with the goal of delaying or preventing progression to metastatic disease. These advances have significantly reshaped the personalized management of patients with recurrence after RT [16,17].

Even in patients who progress to metastatic stages, PC survival can be prolonged for several years, especially with the incorporation of novel hormonal agents such as androgen-receptor pathway inhibitors [18,19]. Nevertheless, early detection remains a cornerstone in optimizing long-term outcomes and guiding treatment decisions.

In this study, we retrospectively analyzed a cohort of patients with localized PC—without lymph-node involvement or distant metastasis—who underwent definitive treatment with EBRT ± ADT. Our objective was to identify clinical and pathological factors associated with BCR. We evaluated both traditional prognostic indicators and histopathological features obtained from diagnostic biopsies, as we believe these samples may hold more prognostic information than is currently incorporated into radiotherapy-based treatment decisions. Our aim was to explore whether integrating such data could improve risk-group stratification.

Furthermore, we assessed the patterns of recurrence, focusing on the anatomical sites and timing of recurrence events throughout follow-up, with the goal of providing evidence to optimize long-term surveillance strategies.

## 2. Materials and Methods

### 2.1. Patient Selection

We conducted a retrospective cohort study, including 629 patients diagnosed with localized PC and treated exclusively with EBRT, with or without ADT (Luteinizing hormone-releasing hormone [LHRH] agonists), between December 2013 and December 2019 at a single institution. After applying the exclusion criteria, which required a minimum follow-up of three years and at least four post-treatment PSA assessments, a total of 444 patients were included in the final analysis (Figure 1). All cases were reviewed by a multidisciplinary tumor board, and written informed consent was obtained from all participants. The study was conducted in accordance with the principles of the Declaration of Helsinki and was approved by the Clinical Research Ethics Committee of Hospital Universitario de Terrassa (approval code: 02-21-100-020). The study was also registered under ClinicalTrials.gov identifier NCT06092918.

### 2.2. Risk Stratification

Patients were categorized into three risk groups based on established criteria:

Low-risk: International Society of Urological Pathology (ISUP) Grade 1, PSA < 10 ng/mL, and clinical stage T1–T2a.

Intermediate-risk: ISUP Grade 2–3, PSA 10–20 ng/mL, or clinical stage T2b–T2c.

High-risk: ISUP Grade 4–5, PSA > 20 ng/mL, or clinical stage ≥ T3a.

Patients classified as high-risk—or intermediate-risk with a Gleason Score (GS) pattern of 4 + 3 (ISUP 3)—underwent staging with axial computed tomography (CT) combined with either whole-body bone scintigraphy or a PET/CT scan using [^18F]-choline, in order to exclude pelvic or distant metastases. [^18F]-choline PET/CT and [^68Ga]-PSMA PET/CT were employed at the time of BCR, depending on availability and institutional protocol.

MRI was not uniformly performed during the study period. Between 2013 and 2016, it was limited to patients with abnormal digital rectal examination due to restricted access. From 2017 onward, MRI was routinely performed either before or after biopsy. This accounts for missing MRI-based staging in a subset of patients.

### 2.3. Technical Parameters of Radiotherapy Planning and Treatment

RT planning was conducted using CT-based simulation with 3 mm slice thickness for the accurate delineation of target volumes and organs at risk. Patients were treated with one of three techniques: three-dimensional conformal radiotherapy (3D-CRT), intensity-modulated radiotherapy (IMRT), or volumetric-modulated arc therapy (VMAT), delivered in either normofractionated or hypofractionated schedules. Specifically, 3D-CRT was delivered using six-field plans with 15 MV or a combination of 6 MV and 15 MV photon beams; IMRT was implemented as step-and-shoot plans using five to seven 6 MV fields; and VMAT employed one or two full arcs with 6 MV photons. Image guidance consisted of weekly portal imaging for 3D-CRT and daily cone-beam CT for IMRT and VMAT.

For target volume delineation, when the clinical target volume (CTV) included only the prostate or the prostate and seminal vesicles (with the latter omitted in low-risk patients), an isotropic margin of 10 mm (for 3D-CRT) or 8 mm (for IMRT/VMAT) was used to define the planning target volume (PTV). In cases involving elective pelvic nodal irradiation, an isotropic margin of 7 mm was applied across all techniques. Pelvic lymph nodes were irradiated in all high-risk patients aged under 75 years and in intermediate-risk patients with a predicted nodal involvement risk greater than 15%, as estimated by the Partin nomogram [20].

Regarding dose schedules, hypofractionated regimens consisted of either 2.5 Gray (Gy) per fraction to the prostate—with or without the inclusion of the seminal vesicles—and an optional simultaneous dose of 1.8 Gy to the pelvic lymph nodes, or 3 Gy per fraction to the prostate and seminal vesicles alone, up to a total dose of 60 Gy. Normofractionated regimens were delivered as 2 Gy per fraction to a total dose of 78 Gy.

Treatment planning was performed with Monaco (Elekta AB, Stockholm, Sweden) and image guidance with Elekta X-ray Volume Imaging (XVI; cone-beam CT). Because this retrospective cohort (2013–2020) spans several institutional upgrades, exact version numbers were not uniformly recorded across all cases.

### 2.4. ADT (Androgen Deprivation Therapy)

LHRHa (luteinizing hormone-releasing hormone agonists) were administered according to international guidelines: 6 months for intermediate-risk patients and 18–36 months for high-risk patients, combined with oral antiandrogens during the first 30 days. Treatment began approximately two months prior to CT simulation.

### 2.5. PSA Monitoring and Definitions

Nadir PSA (nPSA) was the lowest level recorded after treatment. BCR was defined according to the Phoenix criteria: a rise of ≥2 ng/mL above the nPSA. PSA levels were monitored at 3, 6, and 12 months post-EBRT, and every 6 months thereafter.

### 2.6. Biopsy Sampling and Core Analysis

All patients included in the cohort had available data on both the number of positive biopsy cores and the total number of cores obtained per prostatic lobe. The distribution and quantification of positive cores were recorded separately for the right and left lobes, as well as globally. For each lobe, the overall mean number of positive cores was calculated by summing the total number of positive cores across all patients and dividing it by the number of patients (*n* = 444). The same procedure was applied to the combined total of both lobes.

To further evaluate tumor burden as a potential prognostic factor, subgroup-specific means were subsequently calculated for patients who developed BCR and for those who did not, and comparisons were made using appropriate statistical tests.

### 2.7. Statistical Analysis

Descriptive statistics were reported as means with standard deviations or medians with interquartile ranges (IQR, Q1–Q3) for continuous variables, and as absolute and relative frequencies for categorical variables. Group comparisons were performed using the Chi-square test or Fisher’s exact test for categorical variables. For continuous variables, Student’s *t*-test was applied when assumptions of the normality and homogeneity of variances were met; otherwise, non-parametric tests were used.

Univariable analyses were first conducted to assess the association between each variable and BCR. Subsequently, a multivariable logistic regression model was constructed to identify independent predictors of BCR. Categorical variables with more than two levels were transformed into dummy variables. Model outputs included regression coefficients (estimates), standard errors, z-values, and *p*-values, indicating the strength and direction of association between each predictor and the outcome. The timing of BCR was summarized descriptively by follow-up year and displayed as a cumulative incidence plot without time-to-event modelling.

All statistical analyses were performed using the Statistical Package for the Social Sciences (SPSS), version 27 (IBM Corp., Armonk, NY, USA). A *p*-value < 0.05 was considered statistically significant.

## 3. Results

Of the 629 men recruited, 185 were excluded because of having fewer than 4 PSA determinations and/or less than 3 years of follow-up.

A total of 444 patients were included in the final analysis. The median age at diagnosis was 71 years (range 51–85), with a median follow-up of 72 months. Overall, 52/444 (11.7%) experienced biochemical recurrence (BCR). Baseline clinical and treatment-related variables (dichotomous/continuous) are summarized in Table 1, and multi-category clinicopathological variables in Table 2.

### 3.1. Predictive Factors of Biochemical Recurrence

No statistically significant differences were observed in age at diagnosis between patients with and without BCR (median: 72 [range: 54–83] vs. 70 years [51–85], respectively; (*p* = 0.355) (Table 1).

PSA levels at diagnosis were significantly higher in patients who developed BCR, with a median PSA of 10.8 ng/mL and a mean of 30.3 ng/mL, compared to a median of 9.3 ng/mL and a mean of 12.9 ng/mL in patients who did not develop recurrence (*p* = 0.045) (Table 1).

Similarly, no significant differences were found regarding the administration of exclusive EBRT, with a BCR rate of 13.3% among patients who did not receive ADT (*p* = 0.189), nor among those treated with ADT, who presented a BCR rate of 12.6% (*p* = 0.36) (Table 1).

PNI was more prevalent among patients with recurrence, with 18 cases of BCR among 103 patients with PNI and 8 cases among 116 patients without it (*p* = 0.036) (Table 1). Likewise, no significant differences were found according to the fractionation schedule (*p* = 0.119) (Table 1).

Regarding the ISUP Grade Group, the cohort included 76 patients with ISUP grade 1, 255 with ISUP grades 2–3, and 113 with ISUP grades 4–5. A significantly higher proportion of high-grade tumors was observed in the BCR group (*p* = 0.011). Similarly, patients with BCR were more frequently classified as high clinical risk (61.5% vs. 42.6%; *p* = 0.035) (Table 2).

Among the 213 patients who underwent MRI-based clinical staging, the majority were classified as cT2 (*n* = 104; 48.8%), followed by cT3a (*n* = 71; 33.3%) and cT3b (*n* = 38; 17.8%). Although cT3b lesions showed a numerically higher rate of BCR (18.4%), the MRI stage did not reach statistical significance in the univariable analysis (*p* = 0.380) (Table 2).

A total of 19 recurrences were recorded among 114 patients treated with the oldest radiotherapy technique (3D-CRT), representing a recurrence rate of 16.7%, the highest among all techniques. This was followed by IMRT with 5 recurrences in 35 patients (14.3%) and VMAT with 28 recurrences in 295 patients (9.5%). However, no statistically significant differences were observed according to the radiotherapy technique employed (*p* = 0.242) (Table 2).

A higher tumor burden was observed in biopsy samples from patients with BCR, both in terms of the mean number of positive cores per lobe and the proportion of positive cores relative to the total number of sampled cores. The proportion of positive cores was consistently higher in the BCR group compared to the non-BCR group across both lobes (57% vs. 41%, respectively) (Table 1). Specifically, in the right lobe, the mean number of positive cores was 2.98 in the BCR group vs. 2.33 in the non-BCR group (*p* = 0.014); in the left lobe, 3.13 vs. 2.36 (*p* = 0.007); and for the total number of positive cores, 6.11 vs. 4.71 (*p* = 0.01). The differences in proportions between groups were also statistically significant (right lobe: *p* = 0.046; left lobe: *p* = 0.048; total: *p* = 0.005) (Figure 2).

In the multivariable analysis, several independent factors remained statistically significantly associated with an increased risk of BCR.

PSA at diagnosis was confirmed as a significant predictor (OR: 1.01; 95% CI: 1.00–1.02; *p* = 0.05), indicating that higher PSA values are associated with an increased probability of recurrence, even after adjustment for other covariates.

Tumor burden in biopsy samples—expressed both as the absolute number and the proportion of positive cores—showed a strong and consistent association with BCR across all models. The number of positive cores in the right lobe (OR: 3.02; 95% CI: 1.25–7.32; *p* = 0.014), in the left lobe (OR: 2.70; 95% CI: 1.15–6.32; *p* = 0.022), and the total number of positive cores (OR: 8.26; 95% CI: 2.36–28.94; *p* = 0.0009) were all independently associated with a higher risk of recurrence.

Likewise, histological grade ISUP ≥ 4 was confirmed as a significant predictor in the adjusted model (OR: 2.25; 95% CI: 1.05–4.84; *p* = 0.036), reinforcing its prognostic value beyond the univariable analysis.

In contrast, the presence of PNI, although previously associated in the univariable analysis, did not reach statistical significance in the multivariable model (OR: 2.58; 95% CI: 1.02–6.81; *p* = 0.234), suggesting that its effect may be mediated or attenuated by other variables included in the model or possibly influenced by missing data. A summary of this analysis is presented in Table 3.

### 3.2. Recurrence Patterns

Among the patients with BCR, imaging studies (mainly PET/CT) were negative in 12 cases, indicating that no visible disease was detected at the time of relapse. In nine patients, recurrence was confined to the prostate only.

A total of nine patients presented with M1a-exclusive disease (extra pelvis node metastases), while six patients had osseous-only metastases, and five showed exclusive pelvic nodal involvement. The remaining 11 patients exhibited a mixed pattern of recurrence, involving two or more of the above sites.

### 3.3. Time of Recurrence

Of the total BCR events, 2 (3.8%) occurred within the first year after radiotherapy; 10 (19.2%) during the second year; 10 (19.2%) during the third; 8 (15.4%) during the fourth; and 12 (23.1%) during the fifth year. Notably, 10 (19.2%) events were diagnosed beyond five years (Figure 3).

## 4. Discussion

In this retrospective study, we analyzed a cohort of 444 patients with localized PC, without nodal involvement or distant metastases at the time of diagnosis, treated with modern RT techniques and standardized protocols regarding the indication and duration of ADT. The main objective was to confirm and/or identify clinical, pathological, and treatment-related factors that may influence the prediction of BCR, with the aim of enabling a more individualized risk stratification within the currently established risk groups. To this end, we explored the prognostic relevance of key variables such as baseline PSA, PNI, GS/ISUP, and the number–percentage of positive biopsy cores, as well as the timing and patterns of recurrence.

### 4.1. Prostate-Specific Antigen (PSA)

In our cohort, higher baseline PSA was associated with an increased likelihood of BCR after EBRT and remained independently associated in the multivariable model (Table 2 and Table 3).

PSA remains one of the fundamental pillars in the diagnosis, risk stratification, and follow-up of prostate cancer. Evidence from both surgical and, importantly, modern dose-escalated radiotherapy cohorts supports its role as a predictor of BCR [21,22].

There is a proportional relationship between PSA levels and intraprostatic tumor burden, which may partially explain its predictive value for BCR observed in our cohort. However, exceptions do occur. Well-differentiated tumors, for instance, may not produce significantly elevated PSA levels, making the histologic grade (ISUP grade group) essential for proper interpretation in such cases [23]. Conversely, elevated PSA levels may sometimes reflect false positives, most often due to lower urinary tract infections or other benign conditions [24].

PSA is a reliable, accessible, and quantifiable biomarker, but its use should always be complemented by prostate biopsy for diagnosis, and by appropriate imaging for staging or evaluation of recurrence. Among the studies assessing PSA as a prognostic factor, a dose-escalated EBRT series from MSKCC reported pretreatment PSA as an independent predictor of BCR, with each 1 ng/mL increase conferring a measurable rise in biochemical failure risk [25].

### 4.2. Perineural Invasion (PNI)

In our cohort, PNI was more frequent among patients who developed BCR; it reached significance in the univariable analysis but did not remain independently associated after multivariable adjustment, suggesting attenuation by co-variables such as tumor burden and grade and the impact of incomplete reporting.

The biological relevance of PNI has been described in various solid tumors, including PC, where the tumor cell infiltration of perineural spaces is thought to facilitate local tumor spread [26]. PNI, defined as the infiltration of tumor cells into or around nerve structures, has been proposed as a potential pathway for local dissemination. The tumor–nerve interaction may also promote a more aggressive microenvironment, thereby enhancing tumor progression. Several mechanistic hypotheses support its association with more aggressive behavior, including enhanced cellular proliferation and survival within the perineural niche [27,28].

Although PNI is an established adverse factor in surgical (prostatectomy) series, its prognostic value after RT remains uncertain: some studies and meta-analyses suggest an association with BCR, but results are inconsistent [10]. This uncertainty may stem from heterogeneous definitions of PNI, variability in biopsy sampling, and limited quantification (e.g., percentage of nerves involved, number of foci).

In our study, PNI was not reported in nearly half of the biopsy reports, limiting the power of the multivariable analysis and reinforcing the need for standardized and systematic PNI documentation in pathology assessments. Importantly, in recent years the pathology departments of the hospitals where these patients were diagnosed have introduced structured reporting templates to ensure that the presence or absence of PNI is always documented. Previously, some pathologists only recorded it when no perineural invasion was identified, which could create confusion and underreporting. With standardized reporting, more robust datasets will become available, allowing a deeper evaluation of the prognostic role of PNI, particularly in distinguishing its potential impact on local recurrence versus nodal relapse—an aspect that remains insufficiently established. This distinction is clinically relevant, as determining whether PNI predicts local versus nodal recurrence could be key in deciding whether to include pelvic lymph node irradiation, especially in intermediate-risk patients.

### 4.3. Positive Core Number–Percentage

The percentage of positive biopsy cores serves as an indirect marker of tumor burden or overall disease extent within the prostate, and, therefore, it is expected that higher values correlate with an increased risk of BCR [29].

In our study, both the absolute number of positive biopsy cores—whether analyzed by lobe or globally—were significantly associated with BCR. To accurately assess the prognostic value of tumor burden, we first computed the global mean number of positive cores per lobe across the entire cohort, and subsequently analyzed the subgroup-specific means according to recurrence status. Patients who developed BCR had a higher mean number of positive cores in both lobes (Table 1).

These results support positive core burden (count and percentage) as a robust prognostic marker after EBRT, providing information beyond traditional indicators such as PSA, grade (ISUP), and clinical T stage. Although not universally incorporated into standard risk models [30], contemporary RT cohorts—including dose-escalated photon/proton series and externally validated nomograms—also report the incremental prognostic value of positive core burden in RT-treated patients [31]. In particular, the Candiolo classifier, externally validated in a large cohort of patients undergoing EBRT, confirmed that incorporating the percentage of positive biopsy cores significantly improved risk stratification compared with the classical D’Amico groups, achieving higher concordance indices for both biochemical and clinical progression [32].

Taken together, our findings and these external validations reinforce the clinical relevance of positive biopsy cores as a prognostic marker, supporting its integration into modern risk stratification frameworks, especially in the context of contemporary EBRT strategies.

### 4.4. Gleason Score (GS)—ISUP Grade

Histological grading based on the GS, introduced in the 1960s, remains the most widely established predictor of prostate cancer aggressiveness [33]. Although the system has undergone multiple refinements over time, its core principle persists: higher GS reflects less differentiated tumor architecture, which is associated with primary tumors that are more difficult to eradicate—either by surgery or RT—and a higher likelihood of distant metastasis [34].

In our cohort, histologic grade (ISUP grade group) was a significant predictor of BCR after definitive RT. In the univariable analysis, ISUP grade groups were associated with BCR risk (Table 2). In the multivariable logistic model, ISUP ≥ 4 remained independently associated with higher odds of BCR compared with ISUP 1–3 (Table 3), reinforcing its prognostic value beyond the univariable analysis. These findings are consistent with large population-based studies and current international guidelines, which identify higher grade groups as a high-risk subset for BCR [35].

### 4.5. Chronology of Recurrence

A particularly noteworthy finding was that 19.2% of BCR occurred more than five years after the completion of RT. This observation reinforces two key points: first, the need for prolonged follow-up, even in patients who remain disease-free during the initial years after treatment; second, it highlights the advantage of PSA as a robust, cost-effective, and widely accessible biomarker for long-term monitoring.

However, we believe that PSA kinetics alone are insufficient and should be interpreted in conjunction with appropriate initial clinical risk stratification, as the likelihood of recurrence is influenced by pre-treatment clinicopathological features and treatment-related variables. This underscores the importance of early identification of patients at higher risk of BCR, enabling targeted surveillance and helping to prevent unnecessary overburdening of clinical services.

### 4.6. Limitations and Future Directions

This study has several limitations, including its retrospective, single-center design, and the incomplete reporting of certain pathological variables—most notably perineural invasion (PNI). Treatment heterogeneity and the absence of systematic molecular profiling (e.g., genomic classifiers such as Decipher/Prolaris/Oncotype DX and targeted sequencing for DNA-repair alterations or homologous-recombination deficiency), which might further refine risk stratification, limit the generalizability of the findings.

As a future step, we aim to construct a predictive model for biochemical recurrence incorporating the clinical and pathological variables found to be independently associated in this cohort. The prospective validation and integration of emerging biomarkers will be essential to refine risk stratification and support personalized follow-up strategies.

## 5. Conclusions

Biochemical recurrence in localized prostate cancer after radiotherapy was independently associated with higher PSA at diagnosis, an increased proportion of positive biopsy cores, and ISUP grade group ≥ 4 (Gleason score ≥ 8). Lower Gleason categories and perineural invasion showed only a trend toward significance in the multivariable analysis, suggesting that their predictive effect may be attenuated by other clinical covariates or limited by reporting variability. These findings highlight the importance of incorporating detailed biopsy data into recurrence risk assessment. The identification and understanding of these prognostic factors can guide decision-making prior to treatment and support individualized follow-up strategies in patients undergoing curative-intent radiotherapy for prostate cancer.

## Figures and Tables

**Figure 1 biomedicines-13-02185-f001:**
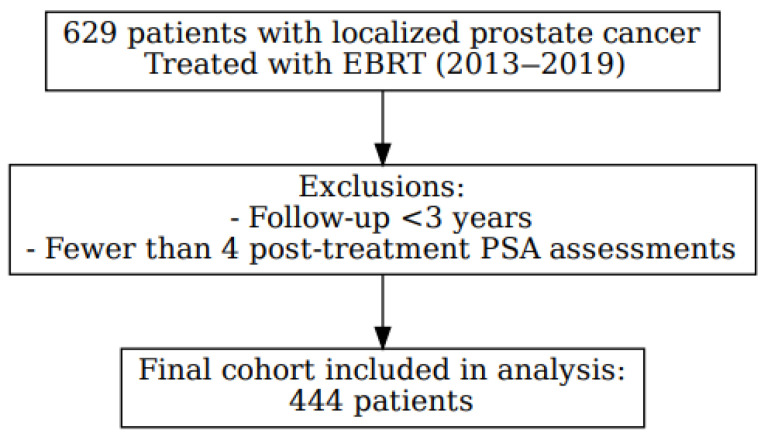
Flowchart of patient selection.

**Figure 2 biomedicines-13-02185-f002:**
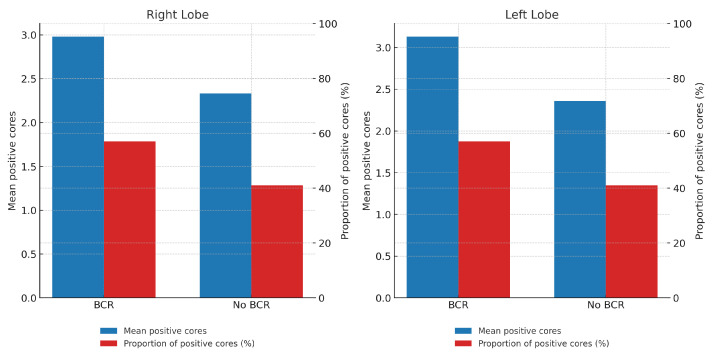
Tumor burden by lobe and BCR status. Bars in blue represent the mean number of positive cores, while bars in red represent the proportion (%) of positive cores in each lobe.

**Figure 3 biomedicines-13-02185-f003:**
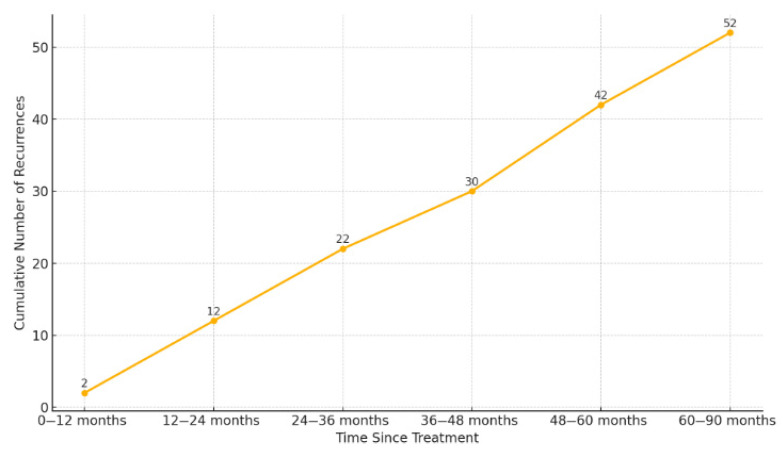
Cumulative incidence of biochemical recurrence over time. Footnotes: The figure shows the number of recurrences accumulated at different time intervals after radiotherapy.

**Table 1 biomedicines-13-02185-t001:** Baseline characteristics of the cohort (*n* = 444). Univariable analysis. Dichotomous and continuous variables.

Characteristic	BCR (*n* = 52)	No BCR (*n* = 392)	*p*-Value
Age, median (range)	72 (54–83)	70 (51–85)	0.355
PSA at diagnosis,			0.045
median/mean (ng/mL)	10.8/30.3	9.3/12.9	
Exclusive Radiotherapy	11 (13.3%)	110 (86.7%)	0.189
Androgen Deprivation Therapy	41 (12.6%)	282 (77.4%)	0.36
Fractionated			0.119
Normofractionated	11 (16.5%)	56 (83.5%)	
Hypofractionated	41 (10.9%)	336 (89.1%)	
Perineural Invasion			0.036
Yes	18 (17.4%)	85 (82.6%)	
No	8 (6.4%)	116 (93.6%)	
Not reported ^a^	26 (11.9%)	191 (88.1%)	
Right Lobe			
Mean positive biopsy cores ^b^	2.98	2.33	0.014
Positive biopsy core percentage (%) ^b^	57%	41%	0.046
Left Lobe			
Mean positive biopsy cores ^c^	3.13	2.36	0.007
Positive biopsy core percentage (%) ^c^	57%	41%	0.048
Both Lobes			
Mean positive biopsy cores ^d^	6.11	4.71	0.01
Positive biopsy core percentage (%) ^d^	57%	41%	0.005

Footnotes: Values are shown as *n* (%) unless otherwise indicated. For continuous variables, medians (range) and/or means are reported as specified. *p*-values are derived from Chi-square or Fisher’s exact tests for categorical variables, and from Student’s t-test or Mann–Whitney U test for continuous variables, as appropriate. ^a^ Percentages are calculated using the number of cases with available data for the corresponding variable. “Not reported” indicates missing pathology information in the original biopsy report. ^b^ Right lobe percentages/means are based on 1074 positive cores out of 2556 sampled cores across 444 patients. ^c^ Left lobe percentages/means are based on 1094 positive cores out of 2563 sampled cores across 444 patients. ^d^ Totals across both lobes are based on 2168 positive cores out of 5119 sampled cores across 444 patients. Abbreviations: BCR, biochemical recurrence; PSA, prostate-specific antigen.

**Table 2 biomedicines-13-02185-t002:** Baseline characteristics of the cohort (*n* = 444). Univariable analysis. Multi-category variables.

Characteristic	BCR, *n* (% of Total) = 52	No BCR, *n* (% of Total) = 392	Total, *n* (% of Total) = 444	BCR Rate Within Group	*p*-Value
Risk Group ^a^					0.035
Low	4 (0.9%)	39 (8.8%)	43 (9.7%)	9.3%	
Intermediate	16 (3.6%)	186 (41.9%)	202 (45.5%)	7.9%	
High	32 (7.2%)	167 (37.6%)	199 (44.8%)	16.1%	
ISUP Grade Groups					0.011
Grade 1	6 (1.4%)	70 (15.8%)	76 (17.1%)	7.9%	
Grade 2–3	24 (5.4%)	231 (52.0%)	255 (57.4%)	9.4%	
Grade ≥ 4	22 (5.0%)	91 (20.5%)	113 (25.5%)	19.5%	
MRI Clinical T Stage Not done ^b^	30 (6.7%)	201 (45.3%)	231 (52.0%)	12.9%	0.380
cT2	10 (2.3%)	94 (21.2%)	104 (23.4%)	9.6%	
cT3a	5 (1.1%)	66 (14.9%)	71 (16.0%)	7.0%	
cT3b	7 (1.6%)	31 (7.0%)	38 (8.6%)	18.4%	
Radiotherapy Technique					0.242
3D-CRT	19 (4.3%)	95 (21.4%)	114 (25.7%)	16.7%	
IMRT	5 (1.1%)	30 (6.8%)	35 (7.9%)	14.3%	
VMAT	28 (6.3%)	267 (60.1%)	295 (66.4%)	9.5%	

Footnotes: For each category, the table shows the number of patients with BCR, without BCR, and the total, together with the within-group BCR rate (BCR/total for that category). *p*-values correspond to overall Chi-square tests across categories. ^a^ Risk groups and ISUP grade groups are defined in Section 2.2. ^b^ MRI Clinical T stage—Not done reflects that MRI was not uniformly performed between 2013–2016 due to access limitations; from 2017 onward, MRI was routinely obtained (see Section 2.2). Abbreviations: BCR, biochemical recurrence; MRI, magnetic resonance imaging; 3D-CRT, three-dimensional conformal radiotherapy; IMRT, intensity-modulated radiotherapy; VMAT, volumetric-modulated arc therapy.

**Table 3 biomedicines-13-02185-t003:** Univariable and multivariable analysis.

Variable	*p* Univariable	Odds Ratio (OR)	95% CI	*p* Multivariable
PSA	0.045	1.01	1.00–1.02	0.05
Positive right-side cores ^a^ (count/proportion)	0.014/0.046	3.04	1.25–7.32	0.014
Positive left-side cores (count/proportion)	0.007/0.048	2.69	1.15–6.32	0.022
Total positive cores (count/proportion)	0.01/0.005	8.25	2.36–28.94	0.009
ISUP grade distribution (ISUP ≥ 4) ^b^	0.011	2.25	1.05–4.84	0.036
Perineural invasion	0.036	2.58	1.02–6.81	0.234

Footnotes: The “*p* univariable” column reports the *p*-value from the univariable test for each predictor (Chi-square/Fisher’s exact for categorical variables; t/non-parametric test for continuous variables). Adjusted odds ratios (OR) with 95% confidence intervals (CI) and corresponding “*p* multivariable” values are derived from the multivariable logistic regression model. ^a^ Core burden was evaluated both as count and as proportion at the univariable level; to avoid collinearity, the multivariable model retained the count-based core variables (right, left, and total positive cores). ^b^ ISUP grade distribution was coded as ISUP ≥ 4 versus ISUP 1–3 in the multivariable model. Abbreviations: OR, odds ratio; CI, confidence interval; ISUP, International Society of Urological Pathology; PSA, prostate-specific antigen.

## Data Availability

The data presented in this study are available on request from the corresponding author. The data are not publicly available due to privacy or ethical restrictions.

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
