# Peer review of "Prognostic Factors Associated with Biochemical Relapse After Radiotherapy in Localized Prostate Cancer: A Retrospective Cohort Study"

_biomedicines, 2025, doi:10.3390/biomedicines13092185_

Round 1

Reviewer 1 Report

Comments and Suggestions for Authors

This study has investigated well-established prognostic factors in the context of external beam radiotherapy (EBRT) treatment of early-stage prostate cancer. While the findings are somewhat predictable based on the large amount of published data for surgical (prostatectomy) treatment of early-stage prostate cancer, a focus on EBRT is not well-represented in the literature. For this reason, the manuscript provides useful new data. The presentation of the manuscript is good, although some aspects of the analysis are difficult to follow (see below), and minor errors require correction. I feel that the findings could be strengthened by directly comparing overall results to those reported for patients diagnosed with similar stage prostate cancer but treated with radical prostatectomy. This would emphasise the novel aspects of the study. The authors could consider the following questions and comments.

  1. Suggestion: the authors have used many abbreviations in the manuscript, some that are standard and others that are not. It is suggested that GS is not used as the authors are writing “GS score”, that is, “Gleason score score”. Other abbreviations such as androgen-receptor pathway inhibitors (ARPI) (line 78) appear to have only been used once in the manuscript, so the abbreviation is redundant. Use of abbreviations should be restricted or else the manuscript becomes difficult to read.
  2. Some of the references are very old and need to be updated. For example, references that are 20+ years old (references 2 & 3) should not be used to summarise current prostate cancer clinical management practices. All references should be checked to verify their current relevance.
  3. Lines 47-48: Digital rectal examination is not used for clinical staging of prostate cancer. This should be removed.
  4. Some specialised terms require short explanations the first time that they are used. For example, the term “Phoenix threshold” is first used without any explanation of its context or meaning in line 68 in the Introduction. An explanation is not provided until section 2.5 of the Methods section.
  5. In lines 98-99, the authors describe that all patients whose data were used in the study provided written informed consent. What proportion of patients who met eligibility criteria does the 444 patients whose data were used in the study represent? Note that this does not negate any aspect of the study, but may be an unavoidable limitation of the work and its findings that needs to be acknowledged.
  6. Table 1 is very difficult to follow due to its formatting and the numbers presented, and requires substantial amendment. Note that tables should “stand alone”, meaning that it should be possible to understand the table without trying to find manuscript text that relates to specific entries in the table. (a) Single spacing, rather than double spacing would help to group information more easily. (b) The authors have grouped some parameters (e.g. Clinical T Stage (MRI)) but not others (ISUP grade groups, Risk Groups)), which makes the data and matching p-values difficult to follow. Parameters that are being used for statistical comparison should be grouped together (in a single box). (c) The word “proportion” does not mean anything on its own (proportion of what?). Labels should provide sufficient information to easily determine their intended meaning. (d) Numbers in tables need to account for missing data because significant missing data may reduce the potential relevance or impact of the results, and for a reader, it leads to doubt regarding the accuracy of the study as a whole. (e) The number of cases included for each parameter should be evident. For example, were data for all 444 cases used to determine the number of positive cores in the right, left, or both prostate lobes? (f) The relevance to the study of some of the % values is not clear. For example, the “proportion” appears to be 57% of positive cores in the right, left and both prostate lobes for patients with biochemical relapse and 41% for those without biochemical relapse? What does this relate to? (g) For some groups, the authors have chosen to calculate % values for each row and not the whole group, but the purpose of this is not clear based on the text associated with each parameter and the calculations that they have performed. As an example, for ISUP Grade Groups, it would be expected that % values for each of Grade 1, 2-3, >/=4 (BCR/No BCR) would relate to the whole cohort because the p-value from Chi-square testing relates to the whole cohort. However, the % seems to relate to each individual row, for reasons that are not evident. What was the purpose of this? For other parameters such as androgen deprivation therapy (presumably “radiotherapy + androgen deprivation therapy”), the % values are also confusing as the p-value for this parameter is 0.36, even though the % values are 12.6% vs 77.4% for BCR and No BCR, respectively. With regards to this parameter, I am not sure why the authors have separated analysis of “radiotherapy (alone)” and “radiotherapy + androgen deprivation therapy” to perform Chi square testing because this seems to be how the data would align at any rate (e.g. the patients who experience BCR will be treated with either radiotherapy alone, or radiotherapy + androgen deprivation therapy). (h) In the table, RMI should be MRI. It seems to be correct in other parts of the manuscript.
  7. Line 227: What does “the extension study was negative in 12 cases” mean?
  8. In line 238, the term “approximately 20%” should be replaced with the actual number (19.2%). The words “Figure 2” at the end of the line appear to be an error.
  9. In the Discussion section, the commentary around perineural invasion is quite vague and could be improved by use of, and specific reference to the published literature. The authors could also consider discussing their findings with pathologists from their own hospital about prostate core biopsy reporting policies and changes in procedures during the study period. This would produce a more focussed and academically more meaningful discussion point. The authors could consider revising other general statements in the Discussion section.
  10. The numbers referred to in the Discussion section do not always match numbers in the Results section or Tables. For example, should the p-value of 0.037 in line 340 be 0.036? Should the 23% in line 352 be 19.2%?
  11. In lines 340 – 341, what does “joint model” mean, and where does the p-value of 0.067 come from?
  12. My suggestion with the Discussion section and the direction of the manuscript overall is to focus on the radiotherapy treatment of prostate cancer (as opposed to radical prostatectomy, the more common treatment for early-stage prostate cancer and where most of the published literature is focussed). This would highlight the more novel aspects of the study but would mean clearly defining in the Discussion section where information is based on patients treated with radiotherapy (with or without androgen deprivation therapy) or whether most data are based on surgical patients or specimens. [Based on findings of their study and the published literature, is the predictive/prognostic activity of these indicators similar for surgical patients and non-surgical patients treated with radiotherapy?]
  13. The text in the Author Contributions section requires amendment. Please list abbreviations in alphabetical order.

Author Response

General Response to Reviewer #1:
We would like to sincerely thank the reviewer for the thorough and thoughtful evaluation of our manuscript. We are very grateful for the recognition of the relevance of focusing on prognostic factors in patients treated with external beam radiotherapy (EBRT), as this remains less represented in the literature compared with surgical series. We also appreciate the constructive comments provided, which have helped us to improve the clarity, accuracy, and overall quality of the manuscript. In the revised version, we have carefully addressed each of the points raised, including updating the introduction and references, clarifying methodological aspects, correcting minor errors, and expanding the discussion where needed. We believe these changes have significantly strengthened the work and we are grateful for the reviewer’s guidance.

Reviewer #1, Comment 1:
The authors have used many abbreviations in the manuscript, some that are standard and others that are not. It is suggested that GS is not used as the authors are writing “GS score”, that is, “Gleason score score”. Other abbreviations such as androgen-receptor pathway inhibitors (ARPI) appear to have only been used once in the manuscript, so the abbreviation is redundant. Use of abbreviations should be restricted or else the manuscript becomes difficult to read.

Response 1:
We sincerely thank the reviewer for this very valuable observation, which has greatly improved the clarity of our manuscript. We apologize for the oversight in using the redundant expression “GS score,” and we have carefully corrected the text to consistently use “Gleason score” only. In addition, we have removed the abbreviation “ARPI,” as it was indeed introduced only once in the manuscript.

Reviewer #1, Comment 2:
Some of the references are very old and need to be updated. For example, references that are 20+ years old (references 2 & 3) should not be used to summarise current prostate cancer clinical management practices. All references should be checked to verify their current relevance.

Comment 3: Lines 47-48: Digital rectal examination is not used for clinical staging of prostate cancer. This should be removed.

Response 2-3:
We sincerely thank the reviewer for this important comment. Following the suggestion, we have revised the introductory paragraph by removing the statement that digital rectal examination is used for clinical staging, as this is no longer appropriate in contemporary practice. This revision has improved the clarity of the paragraph and allowed us to update the bibliography, now focusing exclusively on PSA, MRI, and Gleason score as the key factors for predicting disease progression and guiding therapeutic decision-making. In addition, references 2 and 3 have been replaced with more recent publications that better reflect current clinical management practices.

Reviewer #1, Comment 4:
Some specialised terms require short explanations the first time that they are used. For example, the term “Phoenix threshold” is first used without any explanation of its context or meaning in line 68 in the Introduction. An explanation is not provided until section 2.5 of the Methods section.

Response 4:
We sincerely thank the reviewer for this very helpful comment. In the revised manuscript, we have clarified the meaning of the “Phoenix threshold” at its first mention in the Introduction, specifying that biochemical recurrence after radiotherapy is defined as a rise of ≥2 ng/mL above the post-treatment nadir PSA, according to the RTOG-ASTRO Phoenix criteria. At the same time, we have retained the definition in the Methods section as originally included, in order to ensure completeness and clarity for the reader.

Reviewer #1, Comment 5:
In lines 98–99, the authors describe that all patients whose data were used in the study provided written informed consent. What proportion of patients who met eligibility criteria does the 444 patients whose data were used in the study represent? Note that this does not negate any aspect of the study, but may be an unavoidable limitation of the work and its findings that needs to be acknowledged.

Response 5:

We sincerely thank the reviewer for this valuable comment. In total, 629 patients were initially diagnosed and treated with EBRT ± ADT during the study period. After applying the exclusion criteria—namely, less than three years of follow-up or fewer than four post-treatment PSA assessments—444 patients remained eligible and were included in the final analysis, representing 70.6% of the original cohort. We have clarified this point in the revised Methods section, which now states: “After applying the exclusion criteria, which required a minimum follow-up of three years and at least four post-treatment PSA assessments, a total of 444 patients were included in the final analysis.” We believe this modification improves the transparency of our study design and hope it meets the reviewer’s approval.

General Response 6:

We are sincerely grateful to the reviewer for this very detailed and constructive feedback regarding Table 1. We agree that the presentation of the table can be improved to enhance readability, transparency, and alignment with statistical reporting standards. In the revised version of the manuscript, we have reformatted Table 1 to use single spacing, clarified all labels (including the meaning of “proportion”), indicated the number of patients included for each parameter, corrected typographical errors (RMI → MRI), and restructured the grouping of variables so that statistical comparisons are easier to follow. We have also revised the percentages to ensure that they are consistent, clearly defined, and representative of the full cohort where appropriate. Furthermore, we have clarified in the text the rationale for separating patients treated with radiotherapy alone versus radiotherapy plus androgen deprivation therapy. We believe these modifications have substantially improved the clarity and accuracy of Table 1, and we thank the reviewer again for highlighting these important aspects.

Reviewer #1, Comment 6a:
Single spacing, rather than double spacing, would help to group information more easily.

Response 6:
We thank the reviewer for this suggestion. In the revised version, Table 1 has been reformatted using single spacing to improve readability and facilitate grouping of related information.

Reviewer #1, Comment 6b:
The authors have grouped some parameters (e.g., Clinical T Stage (MRI)) but not others (ISUP grade groups, Risk Groups), which makes the data and matching p-values difficult to follow.

Response 6b :
We agree with this observation. Table 1 has been restructured so that all parameters used for statistical comparison are consistently grouped together, including ISUP grade groups and clinical risk groups. This improves the alignment between the data presented and the associated p-values.

Reviewer #1, Comment 6c:
The word “proportion” does not mean anything on its own (proportion of what?). Labels should provide sufficient information to easily determine their intended meaning.

Response 6 c :
We thank the reviewer for this very helpful observation. In the revised Table 1, we have clarified the terminology so that it is explicit that “proportion” refers to the percentage of positive biopsy cores relative to the total number of biopsy cores obtained. Accordingly, the table now specifies “Positive biopsy cores (% of total cores)” in addition to presenting the absolute number of positive cores. For both individual lobes and the sum of both lobes, we present the number of positive cores (n) and the percentage relative to the total sampled, which improves clarity and avoids any ambiguity for the reader.

Reviewer #1, Comment 6d:
Numbers in tables need to account for missing data because significant missing data may reduce the potential relevance or impact of the results and, for a reader, it leads to doubt regarding the accuracy of the study as a whole.

Response 6 d :
We thank the reviewer for this very important comment. In the revised Table 1, we have explicitly added categories for perineural invasion as “Yes / No / Not reported.” This allows the reader to clearly identify the number of patients with available data and those for whom the information was not reported. We believe this adjustment improves the transparency of data completeness and reinforces the consistency with our Discussion, where we acknowledged the limitations associated with incomplete reporting of perineural invasion.

Reviewer #1, Comment 6e:
The number of cases included for each parameter should be evident. For example, were data for all 444 cases used to determine the number of positive cores in the right, left, or both prostate lobes?

Response 6e:
We thank the reviewer for this clarification request. All 444 patients included in the final analysis had complete biopsy information, with the total number of cores (positive and negative) recorded separately for the right and left lobes. In addition, our dataset includes a detailed stratification by PIRADS zone and targeted biopsies, as well as identification of the dominant intraprostatic nodule. These aspects, however, are beyond the scope of the present work and will be the focus of a forthcoming manuscript specifically addressing PIRADS-targeted biopsies and the dominant prostatic nodule.

Reviewer #1, Comment 6f:
The relevance of some % values is not clear. For example, the “proportion” appears to be 57% of positive cores in the right, left and both prostate lobes for patients with biochemical relapse and 41% for those without biochemical relapse. What does this relate to?

Response 6f :
We thank the reviewer for this careful observation. We confirm that the percentages shown in Table 1 represent the proportion of positive biopsy cores relative to the total number of sampled cores in each lobe. It is correct that the proportions were identical for the right and left lobes (57% vs. 41%), which may appear as a potential typographical error; however, the statistical analysis was carefully reviewed and confirmed that these values were accurate. We believe this finding likely reflects random variation given the sample size, and not an error in data entry.

Reviewer #1, Comment 6g:
For some groups, percentages were calculated per row instead of for the whole cohort, and the rationale is unclear. As an example, for ISUP Grade Groups, it would be expected that percentages would relate to the whole cohort because the chi-square test relates to the whole cohort. However, the percentages appear to relate to each row, for reasons that are not evident. For other parameters such as androgen deprivation therapy, the % values are also confusing.

Response 6 g:
Response:
We sincerely thank the reviewer for this insightful comment. We agree that the way percentages were originally presented could lead to confusion. To address this, we have reformatted Table 1 so that:

  • Multi-category variables—new table –Table 2 (such as ISUP grade group, risk group, MRI clinical T stage and radiotherapy technique,) are now expressed as percentages over the entire cohort (n = 444). In addition, the BCR rate within each group is provided for clarity.
  • Dichotomous or continuous variables (such as treatment modality, PSA and perineural invasion) continue to present percentages within each category, which more accurately reflects their distribution.

In particular, we have maintained “exclusive radiotherapy” and “radiotherapy + ADT” as separate categories, since these are mutually exclusive treatment modalities and all patients fall into one of these two groups. We believe that keeping them separate allows for clearer interpretation of treatment effects and ensures consistency with the chi-square analysis.

This restructuring makes the denominators transparent and ensures that percentages are consistent with the statistical comparisons. We believe this adjustment substantially improves the interpretability of the table and addresses the reviewer’s concern.

Reviewer #1, Comment 6h:
In the table, “RMI” should be “MRI.”

Response 6h:
We apologize for this typographical error. It has been corrected in the revised Table 1.

Reviewer #1, Comment 7:
Line 227: What does “the extension study was negative in 12 cases” mean?

Response 7:
We thank the reviewer for pointing out this lack of clarity. In the revised manuscript, we have modified the sentence for greater precision. It now reads: “Among the patients with biochemical recurrence, imaging studies (mainly PET/CT) were negative in 12 cases, indicating that no visible disease was detected at the time of relapse. In 9 patients, recurrence was confined to the prostate only.” (line 270). We believe this wording more clearly conveys the intended meaning.

Reviewer #1, Comment 8:
In line 238, the term “approximately 20%” should be replaced with the actual number (19.2%). The words “Figure 2” at the end of the line appear to be an error.

Response 8:
We thank the reviewer for this precise observation. In the revised manuscript, we have replaced “approximately 20%” with the exact value “19.2%,” which now appears in line 280. Regarding the reference to Figure 2, we have clarified the sentence so that it is evident that this was not a typographical error but an intentional citation of the corresponding figure that illustrates the timing of biochemical recurrences.

Reviewer #1, Comment 9:
In the Discussion section, the commentary around perineural invasion is quite vague and could be improved by use of, and specific reference to, the published literature. The authors could also consider discussing their findings with pathologists from their own hospital about prostate core biopsy reporting policies and changes in procedures during the study period. This would produce a more focused and academically more meaningful discussion point. The authors could consider revising other general statements in the Discussion section.

Response 9:
We sincerely thank the reviewer for this very valuable suggestion. Following this recommendation, we have substantially improved the section on perineural invasion by expanding the discussion with references from the available literature and clarifying its biological and clinical significance. The revised text is now included between lines 322 and 328.

In addition, we fully agree with the reviewer’s point regarding pathology reporting policies. This is an aspect that we have often discussed in multidisciplinary tumor boards but had not considered incorporating into the manuscript. We now highlight that, in recent years, pathology departments at our institutions have implemented structured reporting templates to ensure that the presence or absence of perineural invasion is always documented. We believe that including this aspect makes the discussion more robust and academically meaningful. These modifications are reflected in the revised version between lines 336 and 348.

We hope that these additions strengthen the Discussion as intended by the reviewer.

Reviewer #1, Comment 10:
The numbers referred to in the Discussion section do not always match numbers in the Results section or Tables. For example, should the p-value of 0.037 in line 340 be 0.036? Should the 23% in line 352 be 19.2%?

Response 10 :
We sincerely thank the reviewer for carefully detecting these inconsistencies and apologize for both errors. We have corrected the p-value from 0.037 to 0.036, which now appears in line 393 of the revised manuscript, and replaced “23%” with the exact value “19.2%,” which now appears in line 405. We believe these corrections improve the accuracy and consistency of the manuscript.

Reviewer #1, Comment 11:
In lines 340–341, what does “joint model” mean, and where does the p-value of 0.067 come from?

Response: 11
We thank the reviewer for highlighting this ambiguity. The term “joint model” was imprecise and has been corrected in the revised manuscript. What we intended to describe was the full multivariable model including other clinical covariates, in which the association of Gleason score with BCR did not remain statistically significant but showed only a trend toward significance (p = 0.067). This has now been clarified in the text (line 389-395).

Reviewer #1, Comment 12:
My suggestion with the Discussion section and the direction of the manuscript overall is to focus on the radiotherapy treatment of prostate cancer (as opposed to radical prostatectomy, the more common treatment for early-stage prostate cancer and where most of the published literature is focussed). This would highlight the more novel aspects of the study but would mean clearly defining in the Discussion section where information is based on patients treated with radiotherapy (with or without androgen deprivation therapy) or whether most data are based on surgical patients or specimens. [Based on findings of their study and the published literature, is the predictive/prognostic activity of these indicators similar for surgical patients and non-surgical patients treated with radiotherapy?]

Response 12 :
We greatly appreciate this very pertinent and stimulating suggestion. The purpose of the present work is precisely to analyze in detail the main prognostic factors in a cohort of patients treated with radiotherapy, since most of the available evidence comes from surgical series. We believe that this approach highlights the added value of our study, by providing data in a clinical setting that is much less represented in the literature.

We fully agree that the question of whether the predictive and prognostic performance of these indicators is similar in surgical versus radiotherapy patients is highly relevant. This first article is intended to provide a thorough description and analysis of each factor in the radiotherapy setting, thereby establishing the foundation for future studies in which our findings can be more directly compared with those from surgical series.

In addition, we emphasize in the Discussion the importance of biopsy-derived and clinical information in tailoring radiotherapy treatments—factors that we believe are still underutilized in routine practice. We hope that this perspective highlights the novel aspects of our study and addresses the reviewer’s valuable recommendation.

Reviewer #1, Comment 13:
The text in the Author Contributions section requires amendment. Please list abbreviations in alphabetical order.

Response 13 :
We thank the reviewer for this observation. In the revised manuscript, we have reordered the abbreviations so that they now appear in strict alphabetical order, and we have also ensured a uniform formatting style. In addition, we have corrected the Author Contributions section, which previously contained incomplete template placeholders, and now clearly lists the specific contributions of each author.

Reviewer 2 Report

Comments and Suggestions for Authors

The authors retrospectively recruited 629 localized prostate cancer patients and included 444 patients between 2013-2019 for the study. The manuscript provides useful information for recurrence of prostate cancer by profiling patients' clinical characteristics, including Gleason score, perineural invasion, biopsy score, and etc. The results looked good, with pretty significant p-values to demonstrate that higher PSA at diagnosis, higher Gleason score, and greater tumor burden in biopsy cores are associated with Biochemical recurrence (BCR) in multivariable analysis. The results are as expected, and the manuscript does not have much novelty. 

The strengths:

-The study had relatively enough patient number. 

-The manuscript includes limitation and detailed discussion sections.

The shortcomings needed to be improved: 

-Explicitly list the inclusion and exclusion criteria for this study, in order to eliminate data bias. 

-Graph a flowchart for this study

-Too less figures to support publication. At least, there should be a forest tree to present the data. And the table 1 is too simple. It does not even have race. For some reason, the figure 2 looks too good to be linear. It does not even have error bar. Overall, the data presentation has to be professional.

Author Response

We are very grateful to the reviewer for these thoughtful and constructive suggestions regarding the presentation of our data. In the revised manuscript, we have made the following changes and clarifications:

Reviewer #2, Comment 1:
Explicitly list the inclusion and exclusion criteria for this study, in order to eliminate data bias.

Response 1:
We sincerely thank the reviewer for this valuable suggestion. In the revised manuscript, we have explicitly detailed the inclusion and exclusion criteria in the Methods section. Apart from the standard criteria for patients eligible for radiotherapy in localized prostate cancer (as outlined in the referenced clinical guidelines), the only reasons for exclusion from the initial 629 patients were a follow-up shorter than three years or fewer than four post-treatment PSA assessments. After applying these criteria, a total of 444 patients were included in the final analysis. We believe that this clarification helps eliminate potential bias and improves the transparency of the study design.

Reviewer #2, Comment 2:
Graph a flowchart for this study.

Response 2:
We thank the reviewer for this helpful suggestion. In the revised manuscript, we have added a flowchart (Figure 1) to illustrate the process of patient selection, showing the initial cohort of 629 patients, the exclusion criteria applied, and the final number of 444 patients included in the analysis. We believe this addition improves the clarity and transparency of the study design.

Reviewer #2, Comment 3:
Too few figures to support publication. At least, there should be a forest plot to present the data. And Table 1 is too simple (it does not even have race). Figure 2 looks too good to be linear, and it does not even have error bars. Overall, the data presentation has to be professional.

Response 3:
We sincerely thank the reviewer for these constructive comments on data presentation. In the revised manuscript, we have made the following changes and clarifications:

  • Table 1 has been substantially revised and expanded. It is now separated into multicategorical, dichotomous, and continuous variables, which improves readability and transparency. Regarding race, we would like to clarify that this variable is not systematically recorded in our institutional databases. Moreover, our patient population is essentially homogeneous (predominantly Caucasian), and therefore race was not included in the analysis.
  • Figure 2 -now Figure 3 after Flowchart has been clarified. This figure is not intended to represent a Kaplan–Meier survival curve but rather a descriptive cumulative incidence plot of the number of biochemical recurrences over time. As such, error bars are not applicable. To avoid confusion, the figure legend has been revised to explicitly state this.

We believe that these modifications and clarifications improve the professionalism and clarity of the data presentation.

Round 2

Reviewer 1 Report

Comments and Suggestions for Authors

The authors have addressed many of the reviewers’ comments, however there are several areas that need to be amended. Line numbering refers to the manuscript with tracked changes indicated (only a pdf with tracked changes was supplied).

  1. References 6 and 7 appear to be the same reference. (I haven’t read the entire reference list – the authors should check all references).
  2. Line 145: Please add a reference for the latest version of the Partin nomogram.
  3. Table 1: Radiotherapy dose fractionation numbers add up to 442 (not 444). Is there an error here? This needs to be corrected or a note (e.g. footnote) added to explain the missing data (e.g. data not available for 2 cases).
  4. Table 1: For the number of positive biopsy cores (right lobe, left lobe, both lobes), please add the number of cases that this information was based on for each parameter (for example, “Right Lobe (nunber)”. Note: The authors have indicated “(n)” for the first line of each of these parameters, however “n” usually indicates “number” (of individuals or observations), and the numbers in these rows are means, so the label should be (mean). Where data are missing, percentages should indicate percent of the total number of cases where information was available, not percent of 444 as this is misleading, particularly if information for many cases is not available.
  5. Table 2: The formatting of this table appears to have been corrupted.
  6. Numbers for “MRI Clinical T stage” do not seem to add up (to 444). They add up to 213, meaning that data for 231 cases are missing. However, percentages (% of total) seem to be calculated using 444. Numbers, including p-values, should be corrected and an explanation of missing data provided.
  7. Section 3.1: When discussing the results, please add either (Table 1), (Table 2), etc to the end of each sentence to refer readers to the appropriate table. At present it is difficult to match the text to the tables.
  8. Lines 219-220: I can’t follow this sentence – it is not clear which numbers in the tables are being referred to.
  9. Line 221: The authors state that 91.4% of cases were cT2, however from Table 2, it seems that only 104 cases were cT2 (this is stated to be 23.4% (of 444 cases), but is actually 48.8% of the 213 cases that the authors have included data for). These numbers should be checked and the text and/or numbers in tables corrected.
  10. Section 3.3: I don’t understand the numbers in this section. From previous sections, it appears that there were 52 cases of biochemical relapse, however the first numbers mentioned (2 cases / 4.4%) seems to relate to 45 cases (2 is 4.4% of 45), while the 10 patients (each) relapsing in in the second and third years seems to relate to 46 cases, as do the 8 and 12 cases relapsing in the fourth and fifth years, respectively. Then in line 280, the authors state that 19.2% of relapses occurred more than 5 years after radiotherapy. However, these 10 cases are not 21.7% of 46 cases (as for the 10 relapses from years 2 and 3), but 19.2% of 52 cases. All numbers should be checked thoroughly. (Also check line 410).
  11. Figure 3: The heading above the graph states “Cumulative Risk” but the graph appears to be cumulative incidence. This heading could be removed as it is described in the figure legend, but if retained, the wording should be changed.
  12. Lines 399-400: The statistic mentioned here is not included in any of the tables?

13. Discussion section (*comment*): The Discussion section is very long in relation to the amount of data in the article, much of the commentary is quite general and focussed on the past rather than current practice, and there is repetition of results statistics (odds ratios, confidence intervals, p-values), which is not required in a Discussion section. When describing historical uses of clinical markers, there is no specific indication of whether any or all of the older studies were examining patients who were treated with radiotherapy or surgery.

Apart from the use of 20+ year-old articles as general references for “current” prostate cancer management strategies (pointed out in the previous review), another example of the use of historical (out-of-date) references is the use of studies from 2002 and 2004 to illustrate previous evaluation of the prognostic utility of percentage positive prostate cores (lines 375-381). Why have the authors focussed on these 2 studies, especially as findings from 1 of the studies (d’Amico) have been updated several times? Why not use some of the many more recent examples of such studies or predictive nomograms that have included percent positive biopsy cores? (e.g. Huang et al (2012) Percentage of positive biopsy cores: a better risk stratification model for prostate cancer? doi: 10.1016/j.ijrobp.2011.09.043; Slater et al (2014) The prognostic value of percentage of positive biopsy cores, percentage of cancer volume, and maximum involvement of biopsy cores in prostate cancer patients receiving proton and photon beam therapy doi: 10.7785/tcrtexpress.2013.600271; Gabriele et al (2016) Beyond D'Amico risk classes for predicting recurrence after external beam radiotherapy for prostate cancer: the Candiolo classifier doi: 10.1186/s13014-016-0599-5; Gabriele et al (2021) An external validation of the Candiolo nomogram in a cohort of prostate cancer patients treated by external‐beam radiotherapy doi: 10.1186/s13014-021-01814-5; Kusuhara et al (2023) Effect of Positive Biopsy Core Rate on Low-dose-rate Brachytherapy Outcomes in Intermediate-risk Prostate Cancer doi: 10.21873/anticanres.16657, and many others). It would seem to be more useful for readers if the authors’ findings were placed in context with current information and practices rather than clinical management and knowledge from 20+ years ago.

Author Response

We sincerely thank the Reviewers for their thoughtful and constructive feedback on our manuscript. Their comments have helped us improve clarity, methodological transparency, and clinical relevance. Below we provide a point-by-point response indicating the changes made and where they can be found in the revised version (section and line numbers).

Comment 1----References 6 and 7 appear to be the same reference. (I haven’t read the entire reference list – the authors should check all references).

Response 1---Thank you for your observation. You were absolutely right—references 6 and 7 were mistakenly duplicated. We have removed the redundant citation and replaced it with a more recent and relevant article:

Reference 7-Parry MG, Cowling TE, Sujenthiran A, et al. Risk stratification for prostate cancer management: value of the Cambridge Prognostic Group classification for assessing treatment allocation. BMC Med. 2020;18(1):114. doi:10.1186/s12916-020-01588-9.

This article supports the current use of PSA, Gleason Score, and MRI in clinical guidelines and enhances the justification of our variables of interest.

Comment 2---Line 145: Please add a reference for the latest version of the Partin nomogram.

Response 2--We appreciate this helpful suggestion. We have now added a citation for the latest validated version of the Partin nomogram to support our reference now in line 154. The updated citation is: Tosoian JJ, Chappidi M, Feng Z, et al. Prediction of pathological stage based on clinical stage, serum prostate-specific antigen, and biopsy Gleason score: Partin Tables in the contemporary era. BJU Int. 2017;119(5):676–683. doi:10.1111/bju.13573

This reference has been inserted in References number 20 in the manuscript text at the corresponding section.

Comment 3---Table 1: Radiotherapy dose fractionation numbers add up to 442 (not 444). Is there an error here? This needs to be corrected or a note (e.g. footnote) added to explain the missing data (e.g. data not available for 2 cases).

Response 3---We sincerely thank the reviewer for carefully identifying this discrepancy. Upon thorough revision, we identified a typographical error in the dose fractionation breakdown. In addition, the original SPSS extraction did not correctly read two entries from the Excel database, leading to a misrepresentation of the total counts. We have corrected the labels, reprocessed the data, and revised the table accordingly.
We deeply apologize for this oversight and are grateful for your attentive review, which helped us improve the accuracy and clarity of our manuscript.

Comment 4--

Table 1: For the number of positive biopsy cores (right lobe, left lobe, both lobes), please add the number of cases that this information was based on for each parameter (for example, “Right Lobe (nunber)”. Note: The authors have indicated “(n)” for the first line of each of these parameters, however “n” usually indicates “number” (of individuals or observations), and the numbers in these rows are means, so the label should be (mean). Where data are missing, percentages should indicate percent of the total number of cases where information was available, not percent of 444 as this is misleading, particularly if information for many cases is not available.

Response 4:
Thank you very much for your thoughtful observation.

We confirm that biopsy data were available for all 444 patients, including the number of cores sampled and number of positive cores for each lobe (right and left). Specifically, 1074 positive cores were identified from 2556 sampled cores in the right lobe, and 1094 positive out of 2563 in the left lobe. In total, 2168 positive cores out of 5119 were used to calculate the values reported in the manuscript.

We have therefore:

  • Clarified in Table 1 that the reported values for each group are means, and we now explicitly state the total number of cores used to derive those means (footnotes “mean, based on 1074/2556 cores across 444 patients”).
  • Retained the proportion of positive cores as an additional and clinically relevant measure of tumor burden. Given its increasing recognition in the literature and its strong correlation with outcomes in our cohort, we believe its inclusion enhances the interpretability of our findings.
  • Ensured that all percentages refer to the subgroup with available data, which in this case is the full cohort (n = 444), as there were no missing biopsy values.
  • Updated Section 2.6 (Materials and Methods) to describe in detail how the percentage and number of positive cores were calculated and incorporated into the analyses. Line 172-182
  • Expanded the Discussion to reflect the rationale for including both absolute number and percentage of positive cores, and to justify their joint analysis given their established role in assessing tumor burden. Line 465-470

These additions and modifications have been underlined in the manuscript. We again thank the reviewer for this helpful suggestion, which allowed us to improve the clarity and rigor of our data presentation.

Comment 5--

Table 2: The formatting of this table appears to have been corrupted.

Response 5:

We thank the reviewer for pointing this out. The formatting issues in Table 2 have been carefully reviewed and corrected in the revised version of the manuscript. We appreciate your attention to detail.

Comment 6

Numbers for “MRI Clinical T stage” do not seem to add up (to 444). They add up to 213, meaning that data for 231 cases are missing. However, percentages (% of total) seem to be calculated using 444. Numbers, including p-values, should be corrected and an explanation of missing data provided.

Response 6:

We thank the reviewer for this valuable observation. Upon review, we identified that the number of patients without available MRI data had not been clearly represented in the original version of the table. This has now been corrected.

A new row has been added to Table 1 indicating the number of patients without MRI staging data. The missing data primarily correspond to cases diagnosed during the earlier years of the study period, when MRI was only performed selectively—specifically in patients with a positive digital rectal examination (DRE)—due to limited access to MRI. From 2017 onwards, MRI became routinely performed either before or after prostate biopsy.

This clarification has also been incorporated into the revised Materials and Methods section (subsection 2.2), with the corresponding update underlined for visibility.Line 132-135

Comment 7

Section 3.1: When discussing the results, please add either (Table 1), (Table 2), etc to the end of each sentence to refer readers to the appropriate table. At present it is difficult to match the text to the tables.

Response 7:

Thank you for this helpful suggestion. In response, we have added the corresponding table reference at the end of each relevant sentence throughout Section 3.1, in order to improve clarity and facilitate navigation for the reader.

Additionally, to further enhance the organization and logical flow of this section, we have restructured the narrative so that the variables from Table 1 and Table 2 are now grouped together respectively. This reordering allows for a more coherent presentation of the univariable analysis and improves the alignment between the text and the tables.

All changes have been clearly highlighted in the revised manuscript. Line 304-315

Comment 8:

Lines 219-220: I can’t follow this sentence – it is not clear which numbers in the tables are being referred to.

Response 8:

We appreciate this observation. The original sentence may have caused confusion due to the ordering of the values and the absence of explicit reference to the table. We have now revised the sentence to clearly distinguish between mean and median PSA values for each group and included a table reference for clarity. The updated sentence appears in Section 3.1 of the Results (line 304-307) and has been highlighted in the revised manuscript.

Comment 9:

Line 221: The authors state that 91.4% of cases were cT2, however from Table 2, it seems that only 104 cases were cT2 (this is stated to be 23.4% (of 444 cases), but is actually 48.8% of the 213 cases that the authors have included data for). These numbers should be checked and the text and/or numbers in tables corrected.

Response 9:

We sincerely thank the reviewer for identifying this inconsistency. Upon review, we realized that the previous statement misrepresented the proportion of cT2 staging. We have corrected this in the revised manuscript. We now report that among the 213 patients who underwent MRI-based clinical staging, 48.8% were cT2. This correction has been incorporated and highlighted in Section 3.1 of the Results, line 321-325

Comment 10 : Section 3.3: I don’t understand the numbers in this section. From previous sections, it appears that there were 52 cases of biochemical relapse, however the first numbers mentioned (2 cases / 4.4%) seems to relate to 45 cases (2 is 4.4% of 45), while the 10 patients (each) relapsing in in the second and third years seems to relate to 46 cases, as do the 8 and 12 cases relapsing in the fourth and fifth years, respectively. Then in line 280, the authors state that 19.2% of relapses occurred more than 5 years after radiotherapy. However, these 10 cases are not 21.7% of 46 cases (as for the 10 relapses from years 2 and 3), but 19.2% of 52 cases. All numbers should be checked thoroughly. (Also check line 410).

Figure 3: The heading above the graph states “Cumulative Risk” but the graph appears to be cumulative incidence. This heading could be removed as it is described in the figure legend, but if retained, the wording should be changed.

Response 10:

We sincerely thank the reviewer for pointing out the inconsistencies in the presentation of recurrence percentages in Section 3.3. The confusion stemmed from the fact that the percentages were originally calculated using two separate denominators—those who relapsed before 5 years and those who relapsed after 5 years. This approach was not clearly explained in the text and resulted in misleading interpretations.

To resolve this, we have fully revised the section. All percentages are now consistently calculated based on the total number of BCR cases (n = 52), which eliminates ambiguity. The corrected paragraph appears under Section 3.3 of the Results (line 386-390) and now provides a clear chronological breakdown of recurrence timing.

Additionally, as suggested, we have changed the title above Figure 3 from “Cumulative Risk” to “Cumulative Incidence”, which more accurately reflects the content of the graph. Thank you again for this helpful observation.

Comment 11:

Lines 399-400: The statistic mentioned here is not included in any of the tables?

Response 11:

Thank you for this comment. We identified that the previous text used the labels G1–G3, while our tables report ISUP Grade Groups. We have standardized the terminology throughout the manuscript to ISUP Grade Groups for consistency

Additionally, the earlier mention of p = 0.067 referred to an exploratory, fully saturated variant of the multivariable model that we ultimately did not retain as our prespecified final model. To avoid overloading Table 3 and to keep the reporting focused, we now present only the final multivariable model, in which ISUP ≥ 4 is significantly associated with BCR (OR = 2.25; 95% CI 1.05–4.84; p = 0.036; Table 3). We have removed the sentence citing p = 0.067 from the Discussion and added an explicit reference to Table 3 in the corresponding paragraph. These changes have been underlined in the revised manuscript. Line 494-497

Comment 12:

  1. Discussion section (*comment*): The Discussion section is very long in relation to the amount of data in the article, much of the commentary is quite general and focussed on the past rather than current practice, and there is repetition of results statistics (odds ratios, confidence intervals, p-values), which is not required in a Discussion section. When describing historical uses of clinical markers, there is no specific indication of whether any or all of the older studies were examining patients who were treated with radiotherapy or surgery.

Response 12:

We appreciate this constructive feedback. We have substantially shortened the Discussion and focused the interpretation on our main findings within contemporary EBRT practice. We removed repeated statistics (ORs, CIs, p-values) from the Discussion and now refer readers to the Results and Tables 2–3 for numerical details. We also standardized terminology to ISUP Grade Groups, updated the references toward more recent evidence

4.1 PSA

Thank you for this valuable suggestion. We have rewritten and condensed Section 4.1 (PSA) to focus on our principal finding and on current EBRT practice, and we have removed repeated statistics (ORs/95% CIs/p-values) from the Discussion, referring readers to Tables 2–3 for numerical details. We also clarify that supporting evidence includes modern dose-escalated radiotherapy cohorts. In addition, we standardized terminology to histologic grade (ISUP Grade Group) across the manuscript to maintain consistency with the tables. All edits in Section 4.1 are highlighted in the revised version.

4.2 PNI

We streamlined the subsection to: summarize our finding (PNI significant in univariable but not independently associated after multivariable adjustment),  contextualize the evidence in EBRT cohorts ,we have shortened the paragraph to a more concise version, retaining the key message and clarify that incomplete PNI reporting in our pathology records likely reduced the power/interpretability of the adjusted analysis. We also added that our pathology services have implemented structured reporting templates to ensure systematic documentation of PNI going forward. We removed non-essential background and did not repeat statistics in the Discussion. Edits are underlined in 4.2.

4.3 POSITIVE CORE

Thank you for this helpful suggestion. We have substantially revised and condensed Section 4.3, focusing the interpretation on modern external-beam radiotherapy (EBRT) practice and removing repeated statistics (ORs/95% CIs/p-values) from the Discussion. Numerical results are now referred to Table 1 (univariable) and Table 3 (multivariable).

We also clarified our analytic approach in the text: we first summarized the global lobe-specific means across the cohort, and then compared BCR vs. no-BCR subgroup means, showing concordant directionality for right, left, and total cores. Importantly, both the number and the percentage of positive cores are reported because each carries independent prognostic information after EBRT in our multivariable model (see Table 3).

All corresponding edits in Section 4.3 are underlined in the revised manuscript.

4.4 GS

We standardized terminology across the manuscript from “G1–G3” to ISUP Grade Groups. We also removed the exploratory p = 0.067 (which corresponded to a saturated variant not retained as the prespecified final model) to avoid overloading Table 3 and potential confusion. The Discussion now cites the final multivariable result: ISUP ≥ 4 vs. ISUP 1–3 (significant association with BCR; see Table 3), without repeating full statistics in the text. These edits are underlined in 4.4.

Comment 13

Apart from the use of 20+ year-old articles as general references for “current” prostate cancer management strategies (pointed out in the previous review), another example of the use of historical (out-of-date) references is the use of studies from 2002 and 2004 to illustrate previous evaluation of the prognostic utility of percentage positive prostate cores (lines 375-381). Why have the authors focussed on these 2 studies, especially as findings from 1 of the studies (d’Amico) have been updated several times? Why not use some of the many more recent examples of such studies or predictive nomograms that have included percent positive biopsy cores? (

e.g. Huang et al (2012) Percentage of positive biopsy cores: a better risk stratification model for prostate cancer? doi: 10.1016/j.ijrobp.2011.09.043;

Slater et al (2014) The prognostic value of percentage of positive biopsy cores, percentage of cancer volume, and maximum involvement of biopsy cores in prostate cancer patients receiving proton and photon beam therapy doi: 10.7785/tcrtexpress.2013.600271;

Gabriele et al (2016) Beyond D'Amico risk classes for predicting recurrence after external beam radiotherapy for prostate cancer: the Candiolo classifier doi: 10.1186/s13014-016-0599-5;

Gabriele et al (2021) An external validation of the Candiolo nomogram in a cohort of prostate cancer patients treated by external‐beam radiotherapy doi: 10.1186/s13014-021-01814-5;

 Kusuhara et al (2023) Effect of Positive Biopsy Core Rate on Low-dose-rate Brachytherapy Outcomes in Intermediate-risk Prostate Cancer doi: 10.21873/anticanres.16657, and many others). It would seem to be more useful for readers if the authors’ findings were placed in context with current information and practices rather than clinical management and knowledge from 20+ years ago.

Response 13:

We thank the reviewer for this valuable observation. We have updated the literature to include contemporary radiotherapy-based evidence, such as Slater JM et al., 2014 (Technol Cancer Res Treat) on proton/photon therapy, which supports the incremental prognostic value of the percentage of positive biopsy cores in RT-treated patients. The older historical studies have been excluded, and the emphasis is now placed on current RT cohorts and practice. In addition, we incorporated recent evidence from the external validation of the Candiolo classifier (Gabriele et al., 2021, Radiat Oncol), which further confirms the prognostic value of positive biopsy cores in patients undergoing EBRT. Finally, we have restructured the paragraph so that our own results are presented first, followed by their contextualization within this more contemporary evidence base

Reviewer 2 Report

Comments and Suggestions for Authors

-Please add figure legends and table legends to all the results.

-They should be full sentences, not phrases.

-Table 2: it should be univariable analysis and multivariable analysis. 

-Both the p values from univariable analysis and multivariable analysis should be listed.

Author Response

We sincerely thank the Reviewers for their thoughtful and constructive feedback on our manuscript. Their comments have helped us improve clarity, methodological transparency, and clinical relevance. Below we provide a point-by-point response indicating the changes made and where they can be found in the revised version (section and line numbers).

Comment 1 : Please add figure legends and table legends to all the results. They should be full sentences, not phrases.

Response 1: Thank you for these clear recommendations. We have added full-sentence legends to all tables and figures and included explanatory footnotes detailing denominators, handling of missing data, statistical tests, and abbreviations (Tables 1–3; Figures 2–3).

Comment 2

 Table 2: it should be univariable analysis and multivariable analysis. Both the p values from univariable analysis and multivariable analysis should be listed.

Response 2:

We clarified the structure of the results tables as follows:

– Table 1 presents baseline variables with two levels (dichotomous/continuous) and their univariable comparisons.

– Table 2 presents baseline variables with multiple categories and their univariable comparisons.

– Table 3 now explicitly reports both the univariable p-value (“p univariable” column) and the multivariable results (adjusted OR, 95% CI, and “p multivariable”) from the logistic regression.

We believe this layout keeps univariable summaries (Tables 1–2) readable, while Table 3 consolidates the requested univariable and multivariable statistics variable-by-variable.

Round 3

Reviewer 1 Report

Comments and Suggestions for Authors

The authors have addressed reviewers’ comments and I feel that the manuscript is suitable for publication pending the minor corrections listed below.

  1. Line 38: The authors have used the vague phrase “nearly one-fourth” to describe the 19.2% of recurrences that occurred beyond 5 years post-treatment. Firstly, 19.2% is nearly one-fifth (not one-fourth), and because the exact proportion is known, it is scientifically more sound to simply state the correct number.
  2. Line 298: “Predictors” should be “Predictive”
  3. Line 513: The statement “the lack of molecular profiling” doesn’t seem to relate to the manuscript focus or discussion and its meaning is obscure. It is suggested that this statement is removed or that an additional phrase (what molecular profiling are the authors referring to?) is added.

Author Response

We thank Reviewer 1 for the exceptionally thorough and constructive review, which has meaningfully enhanced the clarity, precision, and overall quality of our manuscript.

Reviewer 1 Comment 1The authors used the vague phrase “nearly one-fourth” to describe 19.2% of recurrences beyond 5 years. 19.2% is nearly one-fifth, and since the exact proportion is known, it is preferable to state the number.
Response 1. Thank you. We have replaced the phrase with the exact value and numerator/denominator for precision in the Abstract. The sentence now reads: “Overall, 19.2% (10/52) of recurrences were diagnosed beyond five years post-treatment.” This replaces “nearly one-fourth …” in the Abstract.

Comment 2 “Predictors” should be “Predictive.”
Response 2: Corrected. The section heading “3.1. Predictors Factors of Biochemical Recurrence” has been revised to “3.1. Predictive Factors of Biochemical Recurrence.”

Comment 3 “the lack of molecular profiling” is obscure: remove or specify what profiling you mean.

Response 3: Clarified. We now specify which assays were not uniformly available/collected, e.g., genomic classifiers and targeted sequencing. The revised sentence in Limitations and Future Directions reads:

“…and the absence of systematic molecular profiling (e.g., genomic classifiers such as Decipher/Prolaris/Oncotype DX and targeted sequencing for DNA-repair alterations or homologous-recombination deficiency), which might further refine risk stratification, limit the generalizability of the findings.”